# The Optimal Power Allocation for Sum Rate and Energy Efficiency of Full-Duplex Two-Way Communication Network

**DOI:** 10.3390/e24040537

**Published:** 2022-04-11

**Authors:** Hengdong Ye, Zhengchuan Chen, Yunjian Jia, Shutong Chen

**Affiliations:** 1School of Microelectronics and Communication Engineering, Chongqing University, Chongqing 400044, China; yhd@cqu.edu.cn (H.Y.); cst@cqu.edu.cn (S.C.); 2State Key Laboratory of Integrated Services Networks (Xidian University), Xi’an 710071, China

**Keywords:** full-duplex two-way communication, power allocation, sum data rate, energy efficiency

## Abstract

Full-duplex (FD) transmission holds a great potential of improving the sum data rate of wireless communication systems. However, the self-interference introduced by the full-duplex transmitter brings a big challenge to enhance the energy efficiency. This paper investigates the power allocation problem in a full-duplex two-way (FDTW) communication network over an OFDM channel, aiming at improving the sum data rate and energy efficiency. We first characterize the sum rate and energy efficiency achieved in a single-carrier FDTW system. The optimal transmit power that achieves the maximal sum data rate is presented. The energy efficiency maximization problem is solved by using fractional programming. Then we further formulate sum rate and energy efficiency maximization problem in a multi-subcarrier FDTW system. In particular, the sub-optimal transmit power allocation which achieves a decent sum rate improvement is found by using a proposed iterative algorithm. By combining the iterative algorithm and fractional programming, we further maximize the energy efficiency of the multi-subcarrier system. With our proposed algorithm, we can easily obtain an optimal transmit power that approximates the global optimal solution. Simulation results show that using the obtained optimal transmit power allocation algorithm can significantly improve the sum rate and energy efficiency in both single-carrier and multi-subcarrier systems.

## 1. Introduction

With the rapidly growing throughput requirements of user equipment (UE) and the ever-increasing number of connected devices, the next major breakthrough in wireless networking is proceeding at full speed. Full-duplex has been well studied because of the potential of improving the spectral efficiency [1,2]. However, the self-interference (SI) obstructs FD from being widely applied in practice [3]. Specifically, analyses presented in [4,5] showed that when the target spectral efficiency (SE) is high, SI deteriorates the performance of energy efficiency sharply. Along with the development of self-interference cancellation technologies, FD wireless transmission gradually becomes feasible and is well positioned to meet the urgent demand on transmission throughput [6]. Recent advances showed a great potential of canceling the SI up to the receiver noise floor [7,8,9]. The authors in [8] mentioned that 78 dB of interference suppression can be achieved by using multiple self-interference cancellation (SIC) techniques. Thus, the FD communication system has a huge application prospect.

Among the study of FD transmission, the FDTW communication system has attracted great attention from researchers. In [10], the authors studied the energy efficiency (EE) of two-way channels operating in the full-duplex mode, where the corresponding optimal power allocation is derived by using fractional programming. To improve the SE, the mode selection problem for FD-enabled two-way D2D communications is investigated in [11], where the authors maximized the system SE under the conditions of satisfying the minimum rate and maximum power constraints of cellular and D2D users in the communication system. These works both considered the transmit power allocation to achieve higher energy efficiency or SE. This is because the transmit power of the node does not only affect its own transmit data rate, but also generates interference to receivers, leading to the decline of received SNR. Therefore, the power allocation strategy of the nodes plays the most critical role in FDTW communication system.

OFDM technology is widely used to reduce frequency selective fading in broadband communication systems, and the combination of FD technology can effectively improve the transmission rate of the system. The authors of [12] designed a resource allocation scheme for a multi-user FDTW system by rationally allocating available power and carriers, which greatly improves the maximum throughput under a single power constraint. The energy efficiency was also discussed in an OFDM cellular network with a FD relay [13]. The authors considered the joint bandwidth sharing and power allocation problem, and solved it as a three-stage Stackelberg game. In the work [14], a joint optimal power allocation and relay selection scheme is proposed to obtain the optimal transmit power and the cooperative relay under the given time switching factor. The authors considered two cases that the relay nodes harvest energy with single antenna and dual antennas.

The authors of [15] applied big data-based Apache Spark to investigate the 5G wireless communication network. Particularly, they studied the Apache Spark suppression filters and proposed the feedback adaptive filtering for relay systems with multiple-input multiple-out (MIMO) antennas. Another work [16] investigated the downlink outage performance of full-duplex (FD) multi-cell networks where interference cancellation is considered at the receivers. In particular, the authors of [16] put forward a practical scheme to cancel the interference when the power of the interference is very large. The energy efficiency of a decode-and-forward (DF) based standard relay channel was considered in Ref. [17], where the power of residual self-interference and the extra circuit power consumption are modeled as linearly increasing with the relay transmission power. The authors of [18] investigated the energy transmission and computation resource allocation problem in a federated learning-oriented wireless communication network. By adjusting the source precoder and the IRS phase-shift matrix, the authors of [19] maximized the system sum rate of a full-duplex MIMO two-way communication system. Specifically, by decoupling a complex nonconvex problem, the authors formed three subproblems for which closed-form solutions can be obtained.

Both OFDM and FD can improve system transmission spectrum efficiency, and they are widely studied in the sixth generation mobile communications. This is because both schemes can significantly improve data rate and energy efficiency. The authors of [20] proposed a two-step method in which the power is optimally assigned to each subcarrier firstly, and then the assigned power is allocated to the two nodes at each subcarrier. Resource allocation in a full-duplex OFDMA wireless network had been discussed by using matching theory in [21]. However, there is no research on combining the two techniques to improve system data rate and energy efficiency for a two-way communication system. In this paper, we focus on improving the sum data rate and energy efficiency of a FDTW system based on OFDM channel. In this FDTW system, the nodes are more tightly coupled in terms of power, which makes the self-interference of the system more complicated. Hence, it is more difficult to efficiently allocate the power such that a decent sum data rate and energy efficiency can be achieved. We first formulate the optimal problem of selecting optimal power allocation strategy to maximize sum data rate and energy efficiency in a FDTW communication system based on OFDM channel, and each transmit node should satisfy the different power constraints. Then, the fractional programming and iteration algorithm are applied to obtain a suboptimal transmit power strategy. In particular, the proposed algorithms can approach the global optimal solution and significantly improve the sum data rate and energy efficiency in the FDTW system with an OFDM channel. The main contributions of this work are summarized as follows:(1)We consider a single-carrier FDTW system where the nodes have different power constraints. In order to investigate the maximization of the sum data rate and EE in this single-carrier FDTW system, we first analyze how the transmit power, carrier gain, and self-interference coefficient affect the sum data rate. We further optimize the non-convex problem about energy efficiency by applying fractional programming.(2)We extend the studies of single-carrier FDTW system to multi-subcarrier FDTW system. We approach the optimal sum data rate of the system by proposing the cooperative iterative multi-subcarrier system algorithm (CMS). A joint fractional programming and iterative cooperative algorithm (FPIC) is proposed to achieve a suboptimal solution of the energy efficiency of multi-subcarrier system.

The rest of this paper is organized as follows. We introduce the system model in Section 2. Section 3 studies the power allocation in the single-carrier case for sum data rate and energy efficiency. Section 4 extends the studies to multi-subcarrier systems. Simulation results are presented in Section 5. Finally, in Section 6 we conclude this paper.

## 2. System Model

Let us consider a FDTW orthogonal frequency division multiplexing communication system consisting of two nodes $N_{1}$ and $N_{2}$, as shown in Figure 1. There are *M* parallel channels in the system, each of which uses a dedicated OFDM subcarrier with bandwidth *W* Hz. Besides this, when M=1 the system is a single carrier communication system with only one channel; when M>1 the system becomes a multi-subcarrier system with *M* subcarriers and each subcarrier is perfectly orthogonal to each other without inter-subcarrier interference.

Each node receives and transmits its signal by using the *M* subcarriers simultaneously, which causes severe self-interference. After SI cancellation, we can obtain the signal for the *i*-th subcarrier as
(1)$$\begin{array}{c} Y_{1,i}=h_{1,i}\sqrt{P_{2,i}^{}}X_{2,i}+g_{1,i}\sqrt{P_{1,i}^{}}X_{1,i}+Z_{i}, \end{array}$$
(2)$$\begin{array}{c} Y_{2,i}=h_{2,i}\sqrt{P_{1,i}^{}}X_{1,i}+g_{2,i}\sqrt{P_{2,i}^{}}X_{2,i}+Z_{i}, \end{array}$$
where $X_{j,i}$ for j=1,2;i=1,2,⋯,M denotes the signal transmitted from node *j* over the *i*-th subcarrier, and the transmit power of node *j* over the *i*-th subcarrier is denoted as $P_{j,i}$ for j=1,2. The $h_{j,i}$ represents the subcarrier gain, and $g_{j,i}$ represents the coefficient of the *i*-th subcarrier from node *j* to node −j. The $Z_{i}$ denotes additive white Gaussian noise received at node *j* over the *i*-th subcarrier. In this work we assume that the two nodes have the same SIC ability in each subcarrier, thus $g_{1,i}=g_{2,i}=g_{i}$. We consider normalized bandwidth *W* Hz for each subcarrier where the data rate of each subcarrier would be equal to the spectral efficiency of that subcarrier. Let $N_{0}$ denote the power density of the noise.

In this paper, we focus on how the transmission power affects energy efficiency of the system with different power constraints.

Firstly, let us formulate the sum data rate. The signal to interference and noise ratio (SINR) for the two nodes over the *i*-th subcarrier are respectively denoted as $\gamma_{1,i}$ and $\gamma_{2,i}$, i.e.,
(3)$$\begin{array}{cc} \; & \gamma_{1,i}=\frac{h_{1,i}\cdot P_{2,i}^{}}{1+g_{1,i}\cdot P_{1,i}^{}},\;\gamma_{2,i}=\frac{h_{2,i}\cdot P_{1,i}^{}}{1+g_{2,i}\cdot P_{2,i}^{}}. \end{array}$$

Let us further denote the sum data rate of the system as $R(P_{1,i},P_{2,i})$. Then it can be computed by using the Shannon formula, $\gamma_{1,i}$ and $\gamma_{2,i}$.
(4)$$\begin{array}{cc} \; & R\left(P_{1,i},P_{2,i}\right)=\sum_{i=1}^{M}W(\log_{2}\left(1+\gamma_{1,i}\right)+\log_{2}\left(1+\gamma_{2,i}\right)). \end{array}$$

Note that both the $\gamma_{1,i}$ and $\gamma_{2,i}$ are strongly related to the transmit power $P_{1,i}$ and $P_{2,i}$.

In the practical scenarios, the energy supply of the device is not a constant, or the energy supply does not sustain the continuous operation of the device with high power consumption. Different devices are limited by different energy constraints and a part of devices has demanding requirements about energy consumption. Both data rate and energy efficiency are important for those energy-constrained scenarios. Hence, we consider that the nodes in the FDTW system are working with different power constraints.

Through the above analyses, we can formulate the sum data rate maximization problem as $P_{1}$:(5)$$\begin{array}{cc} P_{1}:\;\;\;\;\;\;\; & \max_{P_{1,i},P_{2,i}}\;R=\sum_{i=1}^{M}W(\log_{2}\left(1+\gamma_{1,i}\right)\\ \; & \;\;\;\;\;\;\;\;\;\;\;\;\;\;\;\;+\log_{2}\left(1+\gamma_{2,i}\right)).\;\;\;\;\;\;\;\;\;\; \end{array}$$(6)$$\begin{array}{cc} \; & \;\;\;\;s.t.\;\;\;\sum_{i=1}^{M}P_{1,i}\leq P_{1}^{\max}, \end{array}$$(7)$$\begin{array}{cc} \; & \;\;\;\;\;\;\;\;\;\;\;\;\sum_{i=1}^{M}P_{2,i}\leq P_{2}^{\max}. \end{array}$$

We assume that the devices of both nodes have the same power consumption denoted as $P_{c}$. Based on the formulation of *R*, the energy efficiency of a FDTW system which is denoted by η can be expressed as follows:(8)$$\begin{array}{c} \eta =\frac{R}{\sum_{i=1}^{M}\left(P_{1,i}+P_{2,i}\right)+2\cdot P_{c}}. \end{array}$$

Therefore, let us consider the optimal problem of maximizing the EE as $P_{2}$:(9)$$\begin{array}{cc} P_{2}: & \max_{P_{1,i},P_{2,i}}\;\eta =\frac{R}{\sum_{i=1}^{M}\left(P_{1,i}+P_{2,i}\right)+2\cdot P_{c}},\;\;\;\;\;\;\; \end{array}$$(10)$$\begin{array}{cc} \; & \;\;\;\;s.t.\;\;\;\sum_{i=1}^{M}P_{1,i}\leq P_{1}^{\max}, \end{array}$$(11)$$\begin{array}{cc} \; & \;\;\;\;\;\;\;\;\;\;\;\;\sum_{i=1}^{M}P_{2,i}\leq P_{2}^{\max}. \end{array}$$

From $P_{1}$ and $P_{2}$, one can find that both $P_{1,i}$ and $P_{2,i}$ affect the sum data rate and energy efficiency, where are tightly coupled transmit power. In order to indicate how transmit power affects both *R* and η, let us first explore the the optimal problem in single-carrier system, and then extend the study to multi-subcarrier system.

## 3. Single Carrier System

To gain some insights on the optimal power allocation, let us first study the effect of node transmit power in a single-carrier system. Specifically, we assume that the two nodes have the same carrier gain and self-interference coefficient. We remove the subscript of the $h_{j,i}$ and $g_{j,i}$ for notational simplicity in this section. That is, $h_{1,i}=h_{2,i}=h$ and $g_{1,i}=g_{2,i}=g$. Without loss of generality, let us further assume that $P_{1}^{\max}\leq P_{2}^{\max}$.

### 3.1. Sum Data Rate Maximization

For a single-carrier system, it is given that M=1. Then, one can rewrite the optimal problem $P_{1}$ as $P_{R_{s}}$ as follows:$$\begin{array}{cc} P_{R_{s}}: & \max_{P_{1},P_{2}}\;\;R_{s}=\sum_{i=1}^{M}W(\log_{2}\left(1+\frac{h\cdot P_{2}}{1+g\cdot P_{1}}\right) \end{array}$$(12)$$\begin{array}{cc} \; & \;\;\;\;\;\;\;\;\;\;\;\;+\log_{2}\left(1+\frac{h\cdot P_{1}}{1+g\cdot P_{2}}\right)),\;\;\;\;\;\;\; \end{array}$$(13)$$\begin{array}{cc} \; & \;\;\;s.t.\;\;\;\sum_{i=1}^{M}P_{1}\leq P_{1}^{\max}, \end{array}$$(14)$$\begin{array}{cc} \; & \;\;\;\;\;\;\;\;\;\;\sum_{i=1}^{M}P_{2}\leq P_{2}^{\max}. \end{array}$$

To solve $P_{R_{s}}$, let us first analyze the derivative of $R_{s}$ with respect to $P_{1}$ and $P_{2}$, which can be expressed as follows: (15)$$\begin{array}{cc} \; & \frac{\partial R_{s}}{\partial P_{2}}=\\ \; & \frac{hgP_{2}}{gP_{2}\left(hP_{1}+1+gP_{2}\right)}-\frac{hgP_{2}}{\left(hP_{2}+1+gP_{1}\right)\left(1+gP_{1}\right)}, \end{array}$$(16)$$\begin{array}{cc} \; & \frac{\partial R_{s}}{\partial P_{2}}=\\ \; & \frac{hgP_{1}}{gP_{1}\left(hP_{2}+1+gP_{1}\right)}-\frac{hgP_{1}}{\left(hP_{1}+1+gP_{2}\right)\left(1+gP_{2}\right)}. \end{array}$$

For conciseness, let us define the function $\varphi_{1}$ and $\varphi_{2}$: (17)$$\begin{array}{cc} \;\varphi_{1} & =gP_{2}\left(hP_{1}+1+gP_{2}\right)-\left(hP_{2}+1+gP_{1}\right)\left(1+gP_{1}\right)\\ \; & =\left(P_{2}-2P_{1}\right)+g^{2}\left(P_{2}^{2}-P_{1}^{2}\right)-1-hP_{2}\\ \; & =-g^{2}P_{1}^{2}-2P_{1}-1-hP_{2}+g^{2}P_{2}^{2}+P_{2}, \end{array}$$(18)$$\begin{array}{cc} \;\varphi_{2} & =\left(P_{1}-2P_{2}\right)+g^{2}\left(P_{1}^{2}-P_{2}^{2}\right)-1-hP_{1}\;\;\;\;\;\;\;\;\;\;\;\;\;\;\\ \; & =\left(P_{1}-2P_{2}\right)+g^{2}\left(P_{1}^{2}-P_{2}^{2}\right)-1-hP_{1}\\ \; & =-g^{2}P_{2}^{2}-2P_{2}-1-hP_{1}+g^{2}P_{1}^{2}+P_{1}. \end{array}$$

By comparing Equations (Equation 15) and (Equation 17), Equations (Equation 16) and (Equation 18), one can find that $\varphi_{j}$ would determine the monotonicity of the function $\frac{\partial R_{s}}{\partial P_{j}}$, i.e., if $\varphi_{j}<0$, then $\frac{\partial R_{s}}{\partial P_{j}}>0$; if $\varphi_{j}>0$, then $\frac{\partial R_{s}}{\partial P_{j}}<0$. According to Equation (18), it indicates that when $P_{2}>P_{1}$, that is $P_{2}^{2}>P_{1}^{2}$ and $2P_{2}>P_{1}$, then $\varphi_{2}<0$, $\frac{\partial R_{s}}{\partial P_{2}}>0$. This shows that $R_{s}$ increases with $P_{2}$ when $P_{2}>P_{1}$. Let $P_{1}^{*}$ and $P_{2}^{*}$ denote the best transmit power of $N_{1}$ and $N_{2}$ for system. Then we can obtain the conclusion that $R_{s}^{*}=R_{s}\left(P_{1}^{*},P_{2}^{\max}\right)$ when $P_{1}^{\max}<P_{2}^{\max}$.

Next, let us further find the $P_{1}^{*}$ that achieves the largest sum data rate of the system. Recall that $P_{1}\in \left[0,P_{1}^{\max}\right]$. Note that $\varphi_{1}=0$ is a quadratic equation regarding of $P_{1}$; therefore, we can obtain that
(19)$$\begin{array}{c} △=4g^{2}\left(1+gP_{2}^{\max}+{\left(gP_{2}^{\max}\right)}^{2}-hP_{2}-1\right), \end{array}$$
where △ denote the discriminant of quadratic equation of $\varphi_{1}=0$ w.r.t. $P_{1}$. We can find $\frac{\partial \varphi_{1}}{\partial P_{1}^{}}=-2g-2gP_{1}<0$ all the time if $P_{1}\in \left[0,P_{1}^{\max}\right]$. Hence, there are two cases based on the sign of △ : △≥0 or △<0.

In this case, △≥0. Let us denote the positive solution of equation $\varphi_{1}=0$ as $P_{1}^{\prime }$. We can obtain
(20)$$\begin{array}{c} \begin{array}{c} P_{1}^{\prime }=\frac{\sqrt{4g^{2}\left(1+gP_{2}^{\max}+{\left(gP_{2}^{\max}\right)}^{2}-hP_{2}-1\right)}-1}{g}. \end{array} \end{array}$$

On the basis of Equation (Equation 17), one can find that $\varphi_{1}>0$ when $P_{1}\in \left[0,\;P_{1}^{\prime }\right]$, $\varphi_{1}<0$ when $P_{1}\in \left[P_{1}^{\prime },P_{1}^{\max}\right]$. This indicates that $P_{1}^{*}=0$ or $P_{1}^{\max}$. In order to determine the largest sum data rate, we define the function $\chi =R_{s}\left(0,\;P_{2}^{\max}\right)-R_{s}\left(P_{1},\;P_{2}^{\max}\right)$ to identify $P_{1}^{*}$. Through △≥0, we can obtain that
(21)$$\begin{array}{c} h\leq \frac{1+gP_{2}^{\max}+{\left(gP_{2}^{\max}\right)}^{2}}{P_{2}^{\max}}:=h_{0}. \end{array}$$

Let us denote the positive solution of χ=0 as $P_{1}^{\prime \prime }$. Then,
(22)$$\begin{array}{c} P_{1}^{\prime \prime }=\frac{6P_{2}^{\max}+4hP_{2}^{\max}}{2+hP_{2}^{\max}}. \end{array}$$

Analyzing Equation (Equation 22) based on $P_{1}^{\max}<P_{2}^{\max}$ and $0<h\leq h_{0}$, $P_{1}^{\prime \prime }>P_{2}^{\max}>P_{1}^{\max}$ all the time. Therefore, if $h\leq h_{0}$ and $P_{1}^{\max}<P_{2}^{\max}$, then $P_{1}^{*}=0$, and $P_{2}^{*}=P_{2}^{\max}$, and the optimal $R_{s}$ achieves at $R_{s}^{*}=R_{s}\left(0,P_{2}^{\max}\right)$.

In this case △<0, we can obtain the relationship between carrier gain *h* and $h_{0}$:(23)$$\begin{array}{c} h>\frac{1+gP_{2}^{\max}+{\left(gP_{2}^{\max}\right)}^{2}}{P_{2}^{\max}}:=h_{0}. \end{array}$$

Since △<0, the sum data rate increases as $P_{1}$ increases. Therefore, if $h>h_{0}$, then $P_{1}^{*}=P_{1}^{\max}$, and $P_{2}^{*}=P_{2}^{\max}$, and the optimal $R_{s}$ achieves at $R_{s}^{*}=R_{s}\left(P_{1}^{\max},P_{2}^{\max}\right)$.

### 3.2. Energy Efficiency Maximization

In this section, we further maximize the energy efficiency of a single-carrier FDTW system. Specifically, in this case, M=1. Let us obtain the problem degradation as $P_{\eta_{s}}$ from $P_{2}$ for a single-carrier FDTW system as follows:(24)$$\begin{array}{cc} P_{\eta_{s}}:\;\;\;\; & \max_{P_{1},P_{2}}\;\;\eta_{s}=\frac{R_{s}}{P_{1}+P_{2}+2\cdot P_{c}},\;\;\;\;\;\;\; \end{array}$$(25)$$\begin{array}{cc} \; & \;\;\;s.t.\;\;P_{1}\leq P_{1}^{\max}, \end{array}$$(26)$$\begin{array}{cc} \; & \;\;\;\;\;\;\;\;\;\;P_{2}\leq P_{2}^{\max}. \end{array}$$

Note from Equation (Equation 24) that the transmit power occurs in both the numerator and denominator, which indicates that the transmit power of different nodes are coupled with each other and difficult to be solved directly. Hence, let us optimize this non-convex problem by using fractional programming mentioned in [22,23,24]. The optimal solution of $P_{\eta_{s}}$ can be achieved if and only if
(27)$$\begin{array}{c} R_{s}^{*}-\eta_{s}^{*}\cdot P_{j}^{*}=0, \end{array}$$
where $R_{s}^{*}$ denotes the optimal sum data rate, $P_{j}^{*}$ denote the corresponding optimal transmit power, and $\eta_{s}^{*}$ is the maximum energy efficiency of $P_{\eta_{s}}$ [25].

Let us define $F\left(\eta_{s}\right)=R_{s}-\eta_{s}P_{j}$, and we can easily find the optimal EE $\eta_{s}^{*}$ by using bisection search because the function $F(\eta_{s})$ is monotonically decreasing. Furthermore, the equation $F\left(\eta_{s}\right)=R_{s}^{*}-\eta_{s}^{*}\cdot P_{j}^{*}=0$ can be obtained when (Equation 27) follows and the optimal EE is achieved. In each searching step, we solve the following problem to get the optimal transmit power with a initial energy efficiency $\left(\eta_{mid}\right)$.
(28)$$\begin{array}{cc} P_{3}:\;\;\;\;\;\;\;\;\;\;\;\; & \max_{P_{j}}\;F\left(\eta_{s}\right),\;\;\;\;\;\;\;\;\;\;\;\;\;\;\;\;\;\;\;\;\;\;\;\;\;\;\; \end{array}$$
(29)$$\begin{array}{cc} \; & \;\;s.t.\;\;\;P_{j}\leq P_{j}^{\max}. \end{array}$$

The objective function has a convex feasible region and is continuous with respect to $P_{j}$. Therefore, the maximum value of $F(\eta_{s})$ can be found when transmit power meet the criteria (29). In a single-carrier FDTW system, to obtain the optimal transmit power is a convex problem for any given $\eta_{s}$ as an initial $\eta_{mid}$. Hence, by several iterations, we can obtain the optimal transmit power $P_{j}^{*}$ and energy efficiency $\eta_{s}^{*}$ until $F(\eta_{s}^{*})=0$.

We design the Algorithm 1 based on above analyses to search the optimal energy efficiency ($\eta_{s}^{*}$).
**Algorithm 1** Fractional programming algorithm for $P_{\eta_{s}}$**Input:** The initial energy efficiency $\eta_{mid}$ which is defined on the closed interval [a,b], the margin of error δ.**Output:** the optimal energy efficiency $\eta_{s}^{*}$.1:Initializing system parameters.2:j←1.3:Find optimal transmit power $P_{j}^{\mathrm{temp}}$ by standard convex optimization algorithm and get $F^{\max}\left(\eta_{mid}\right)$.4:**if** $F^{\max}\left(\eta_{mid}\right)<0$. **then**5:    a $\leftarrow \eta_{mid}$.6:**else**7:    b $\leftarrow \eta_{mid}$.8:**end if**9:**if** $F^{\max}\left(\eta_{mid}\right)=0$ or $|\eta_{mid}-\left(a+b\right)/2|<\delta$. **then**10:   $\eta_{s}^{*}\leftarrow \eta_{mid},\;P_{j}^{*}\leftarrow P_{j}^{\mathrm{temp}}$, j←2.11:   Go to 3, and take the same steps for the other node.12:**else**13:   $\eta_{mid}\leftarrow \left(a+b\right)/2.$14:   Go to 4.15:**end if**

## 4. Multi-Subcarrier System

In this section, we analyze the sum data rate and EE of a FDTW system with multi-subcarrier. Since the number of subcarrier increases, the optimal power allocation problem becomes more complicated.

### 4.1. Sum Data Rate Maximization

For the multi-subcarrier condition, let us assume that different nodes can share their system information which have been mentioned in the last section. Nodes can send their messages and consider how much impact they made for this system, such as sum data rate, SI. Furthermore, Nodes can choose an appropriate transmit power to achieve a balance between higher transmit data rate and stronger self-interference. The data rate of each node is jointly influenced by the matching full-duplex two-way nodes. We can rewrite the optimization problem $P_{1}$ for this condition as follows:(30)$$\begin{array}{cc} P_{R_{m}}:\;\;\;\;\; & \max_{P_{j,i},P_{-j,i}}\;\;\;R_{m}\left(P_{j,i},\;P_{-j,i}\right),\;\;\;\;\;\;\;\;\;\;\;\;\; \end{array}$$(31)$$\begin{array}{cc} \; & \;\;\;\;\;s.t.\;\;\;\;\sum_{i=1}^{M}P_{j,i}\leq P_{j}^{\max}, \end{array}$$(32)$$\begin{array}{cc} \; & \;\;\;\;\;\;\;\;\;\;\;\;\;\sum_{i=1}^{M}P_{-j,i}\leq P_{-j}^{\max}. \end{array}$$

A node will consider the other’s transmit strategy to select the optimal transmit power, but it does not know the transmit strategy chosen by the other node. Let us first define a Lagrange function to solve the problem based on $P_{R_{m}}$:(33)$$\begin{array}{c} \begin{array}{cc} \; & L_{2}\left(P_{j,i},P_{-j,i},\lambda_{j},\lambda_{-j}\right)=R_{m}\\ & -\lambda_{j}\left(\sum_{i=1}^{M}P_{j,i}-P_{j}^{\max}\right)-\lambda_{-j}\left(\sum_{i=1}^{M}P_{-j,i}-P_{-j}^{\max}\right). \end{array} \end{array}$$

One should note that the solution of $P_{R_{m}}$ is following the equations through Equation (Equation 33), as follows: (34)$$\begin{array}{c} \frac{\partial L_{2}\left(P_{j,i},P_{-j,i},\lambda_{j},\lambda_{-j}\right)}{\partial P_{j,i}}=0, \end{array}$$(35)$$\begin{array}{c} \frac{\partial L_{2}\left(P_{j,i},P_{-j,i},\lambda_{j},\lambda_{-j}\right)}{\partial P_{-j,i}}=0. \end{array}$$

Because Equations (Equation 34) and (35) are transcendental equations, and the transmit power of two nodes are coupled together, it is difficult to obtain the closed form solution as $P_{j,i}^{*}$ of the two nodes’ best transmit power at the same time. We propose a cooperative iterative multi-subcarrier scheme (CMS) to obtain suboptimal $R_{m}$.

We assume that node selects the transmit strategy through the previous decisions of the other node; hence, we can obtain the optimal transmit power in each round iteration process as $P_{j,i}^{\left(n\right)}$ as follows:(36)$$\begin{array}{c} P_{j,i}^{\left(n\right)}={\left[\frac{\sqrt{{b_{j,i}}^{2}-4a_{j,i}\left(c_{j,i}-\lambda_{j}\right)}-b_{j,i}}{2a_{j,i}}\right]}^{+},\forall i. \end{array}$$

In Equation (Equation 36), the parameters are defined as
(37)$$\begin{array}{c} \;\;\left\{\begin{array}{c} a_{j,i}=2\lambda_{j}h_{j,i}g_{i},\\ b_{j,i}=\lambda_{j}h_{j,i}+\lambda_{j}g_{i}+\lambda_{j}{g_{i}}^{2}P_{-j,i}^{\left(n-1\right)},\\ c_{j,i}=\lambda_{j}g_{i}P_{-j,i}^{\left(n-1\right)}+{g_{i}}^{2}P_{-j,i}^{\left(n-1\right)}+\lambda_{j}-h_{j,i}. \end{array}\right. \end{array}$$

We fix transmit power at the beginning and obtain the other node’s transmit power, then repeat the same operation for the other node as shown in Figure 2. We search for an optimal transmit power $P_{1,i}^{\left(1\right)}$ with (Equation 34) satisfied, by given initial transmit power $P_{2,i}^{\left(0\right)}$ of $N_{2}$. By substituting $P_{1,i}^{\left(1\right)}$ into (35), we can obtain a $P_{2,i}^{\left(1\right)}$ which satisfies (35) through using a standard convex optimization algorithm. After that, iteratively substituting $P_{2,i}^{\left(1\right)}$ into (Equation 34), one can obtain a new optimal power for $N_{1}$, $P_{1,i}^{\left(2\right)}$. Then repeat in this way until $|P_{1,i}^{\left(n\right)}-P_{1,i}^{\left(n-1\right)}|<\delta$ and $|P_{2,i}^{\left(n\right)}-P_{2,i}^{\left(n-1\right)}|<\delta$. The suboptimal transmit strategy can be obtained after *n*-round iteration to approach the suboptimal solution as $\left(P_{1,i}^{*},P_{2,i}^{*}\right)$. This iterative algorithm greatly reduces the complexity of the calculation.
**Algorithm 2** Search for the optimal solution of $P_{R_{m}}$**Input:** Maximum transmit power $P_{j}^{\max}$, algorithm total iteration times *N*, the initial transmit power at *i*-th subcarrier of *j*-th node $P_{j,i}^{\mathrm{temp}}$,the margin of error δ.**Output:** optimal transmit power $P_{1,i}^{*}$ and $P_{2,i}^{*}$.1:Initializing system parameters.2:n←1.3:Obtaining the optimal transmit power $P_{1,i}^{\left(n\right)}$ according to Equation (Equation 36).4:**if** 
$|P_{1,i}^{\left(n\right)}-P_{1,i}^{\left(n-1\right)}\left|<\delta \;\&\;\right|P_{2,i}^{\left(n\right)}-P_{2,i}^{\left(n-1\right)}|<\delta$ 
**then**5:   **return** 
$P_{1,i}^{*}\leftarrow P_{1,i}^{\left(n\right)},P_{2,i}^{*}\leftarrow P_{2,i}^{\left(n-1\right)}$6:**end if**7:Obtaining the optimal transmit power $P_{2,i}^{\left(n\right)}$ according to Equation (Equation 36).8:**if** 
$|P_{1,i}^{\left(n\right)}-P_{1,i}^{\left(n-1\right)}\left|<\delta \;\&\;\right|P_{2,i}^{\left(n\right)}-P_{2,i}^{\left(n-1\right)}|<\delta$
 **then**9:   **return** 
$P_{2,i}^{*}\leftarrow P_{2,i}^{\left(n-1\right)},P_{1,i}^{*}\leftarrow P_{1,i}^{\left(n\right)}$10:**else**11:   **if** n<N **then**12:     n←n+1, go to 3.13:   **end if**14:**end if**

We design Algorithm 2 based on these analyses to search the suboptimal solution at $\left(P_{1,i}^{*},P_{2,i}^{*}\right)$. The suboptimal sum data rate could be achieved as $R_{m}\left(P_{1,i}^{*},P_{2,i}^{*}\right)$.

### 4.2. Energy Efficiency Maximization

Through our analyses in Section 3, we have used fractional programming to optimize the single-carrier energy efficiency problem, and the iterative scheme mentioned in the previous section to achieve the optimal sum data rate. We rewrite the optimal problem $P_{2}$ for multi-subcarrier condition as $P_{\eta_{m}}$.
(38)$$\begin{array}{cc} P_{\eta_{m}}:\;\; & \max_{P_{1,i},P_{2,i}}\;\;\eta_{m}=\frac{R_{m}}{\sum_{i=1}^{M}P_{1,i}+P_{2,i}+2\cdot P_{c}},\;\;\;\;\; \end{array}$$
(39)$$\begin{array}{cc} \; & \;\;\;\;s.t.\;\;\;\sum_{i=1}^{M}P_{1,i}\leq P_{1}^{\max}, \end{array}$$
(40)$$\begin{array}{cc} \; & \;\;\;\;\;\;\;\;\;\;\;\;\sum_{i=1}^{M}P_{2,i}\leq P_{2}^{\max}. \end{array}$$

As the number of subcarriers *M* increases, the transmit power between the subcarriers is more tightly coupled, and it becomes more difficult to solve $P_{\eta_{m}}$. Therefore, we propose the combining fractional programming and iterative cooperative algorithm (FPIC). The iterative cooperation scheme is used to reduce the complexity of a single calculation in the multi-subcarrier system, and the non-convex problem of energy efficiency is solved by fractional programming. By using the proposed FPIC scheme, the sub-optimal solution is continuously approached to the optimal solution of the system.

In our proposed FPIC scheme, we first assume that $N_{2}$ allocates equal-power $P_{2,i}^{\mathrm{temp}}$ in each subcarrier for $N_{1}$ in the beginning. The optimal transmit power $P_{1,1}^{\mathrm{temp}}$ and $\eta_{m}^{\mathrm{temp}}$ of the first subcarrier can be found by applying fractional programming with a given initial $\eta_{mid}$. Then, let $\eta_{m}^{\mathrm{temp}}$ be the $\eta_{mid}$ for next subcarrier, we can achieve the $P_{1,2}^{\mathrm{temp}}$ and refresh the $\eta_{m}^{\mathrm{temp}}$. Until all subcarriers of $N_{1}$ are optimized, we note the latest $\eta_{m}^{\mathrm{temp}}$ as $\eta_{m}^{*(1)}$ and $P_{1,i}^{\mathrm{temp}}$ as $P_{1,i}^{*(1)}$. The $N_{2}$ can obtain the transmit power $P_{1,i}^{\mathrm{temp}}$ and the latest $\eta_{m}^{\mathrm{temp}}$. Then, we fix the transmit power of $N_{1}$ to $P_{1,i}^{*(1)}$. Hence, $N_{2}$ can repeat the same optimal progress as $N_{1}$ for each subcarrier to refresh $\eta_{m}^{\mathrm{temp}}$ and transmit power $P_{2,i}^{\mathrm{temp}}$. Until all subcarriers of $N_{2}$ are optimized, we note the latest $\eta_{m}^{\mathrm{temp}}$ as $\eta_{m}^{*(2)}$ and $P_{2,i}^{\mathrm{temp}}$ as $P_{2,i}^{*(2)}$, and substitute them to $N_{1}$. Repeating in this way after *n*-round of iteration until $\eta_{m}^{*(j)}$ is converges, we can achieve the optimal energy efficiency $\eta_{m}^{*}$.

We design Algorithm 3 based on the above analyses to search for the optimal energy efficiency ($\eta_{m}$).
**Algorithm 3** Search for the optimal solution of $P_{\eta_{m}}$**Input:** Maximum transmit power $P_{j}^{\max}$, algorithm total iteration times *N*, the initial transmit power at *i*-th subcarrier of *j*-th node $P_{j,i}^{\mathrm{temp}}$, the initial iteration energy efficiency $\eta_{mid}$.**Output:** 
Energy efficiency $\eta_{m}^{*}$.1:Initializing system parameters.2:n←1,j←1.3:For each subcarrier of *j*-th node, obtaining the optimal transmit power at $P_{j,i}^{\left(n\right)}$ and the temporary optimal energy efficiency $\eta_{m}^{\mathrm{temp}}$ by Algorithm 1.4:$\eta_{mid}\leftarrow \eta_{m}^{\mathrm{temp}}$ and $P_{j,i}^{\mathrm{temp}}\leftarrow P_{j,i}^{*(n)}$5:**if** Computed all subcarrier of *j*-th node **then**6:   $\eta_{m}^{*(j)}\leftarrow \eta_{m}^{temp}$, $\eta_{mid}\leftarrow \eta_{m}^{temp}$, $P_{j,i}^{*(j)}\leftarrow P_{j,i}^{temp}$.7:   j←2, go to 3.8:**end if**9:**if** 
n<N 
**then**10:   n←n+1, go to 3.11:**else**12:   $\eta_{m}\leftarrow \eta_{m}^{*(j)}$13:**end if**

## 5. Simulation Results

In this section, we provide some numerical simulations of our proposed theory. Firstly, we analyze and compare the sum data rate of single-carrier system $R_{s}$ under different assumptions of *h*. Then, we analyze the $R_{m}$ among the proposed cooperative iterative multi-subcarrier scheme (CMS), the evenly distributed power multi-subcarrier scheme (EDMS) which be as the baseline, and the non-cooperative iterative multi-subcarrier scheme (NCMS) where each node distributes the power for each subcarrier through the water-filling method. Then, we compare the energy efficiency of the multi-subcarrier FDTW system when applying different schemes, i.e., EDMS, FPIC, and Max Value, which means that the multi-subcarrier system achieves the optimal theoretical value of EE. Finally, we analyze how the $\eta_{m}$ achieved by the proposed FPIC scheme varies with the transmit power of each node.

Figure 3 shows how the sum data rate varies with the transmit power in the single-carrier with different single-carrier gains, e.g., $h=h_{0}/4$, $h=h_{0}/2$, $h=2h_{0}$ or $h=5h_{0}$. When *h* takes a low value, e.g., $h\leq h_{0}$, the $R_{s}$ decreases first along with the increase of $P_{1}$. This manifests that there exists an optimal transmit power that achieves the best sum data rate. Furthermore, the system with higher *h*, e.g., $h=2h_{0}$ or $h=5h_{0}$, can obtain a higher sum data rate. This indicates that higher transmit power is not suitable to achieve a better sum data rate due to strong self-interference and poor single-carrier condition.

Then, we compare the system sum data rate with transmit power between the CMS, NCMS and EMDS which are introduced at the beginning of Section 5 in Figure 4. We can see that both the CMS and NCMS schemes are able to improve the sum data rate of the system compared with the EDMS solution. Furthermore, because the CMS scheme can consider both the data rate gain and the interference caused by both sides when allocating power, it can have a maximum rate improvement of 10% at the same transmitting power.

Next, we compare the $\eta_{m}$ of the FPIC scheme to that of the EDMS scheme in Figure 5. Along with the increase of *N*, the energy efficiency $\eta_{m}$ of FPIC increases rapidly first, then stabilizes when N=6. When the FPIC scheme has optimized the second subcarrier, that is N=2, it can already achieve better energy efficiency than the EDMS scheme.

Finally, we compare the FPIC scheme and the EDMS scheme in Figure 6. In the simulation for $\eta_{m}$ varies with $P_{1}^{\max}$ over different $P_{2}^{\max}$, we fix $P_{2}^{\max}=200$ mW or 300 mW. Along with the increase of $P_{1}^{\max}$, the blue square curve and black circular curve represent the EDMS scheme, where they first gradually reach the optimal $\eta_{m}$ and then gradually decrease. This shows that the blindly random allocation of power will lead to worse interference, which greatly reduces the energy efficiency of the FDTW system. This disadvantage does not occur in the proposed FPIC scheme. For maximizing energy efficiency, the algorithm maintains good consistency in the overflowed transmit power constraint system. The two transmitting nodes continuously adjust the transmission power through iterative cooperation to ensure that the transmit power of the system is always sub-optimal. Even if $N_{1}$ and $N_{2}$ both have sufficient energy supply, and neither party will selfishly increase its transmit power. The iterative and cooperation thoughts of the proposed FPIC scheme are well verified.

## 6. Conclusions

This work discussed the sum data rate and energy efficiency optimal for a full-duplex two-way communication system. We formulated and analyzed the optimization problem of the sum data rate and energy efficiency for single-carrier and multi-subcarrier conditions. Then, with the CMS and FPIC schemes, we achieved the optimal value of *R* and η of nodes in the energy-constrained FDTW system. By comparing these power allocation schemes, it is found that FPIC can improve EE both in single-carrier and multi-subcarrier conditions. Furthermore, under the multi-subcarrier cases, our CMS scheme can greatly increase the sum data rate of the system. Moreover, with our proposed FPIC and CMS schemes, It can easily obtain the optimal transmit power for each communication node in the system without a complicated solving process.

## Figures and Tables

**Figure 1 entropy-24-00537-f001:**
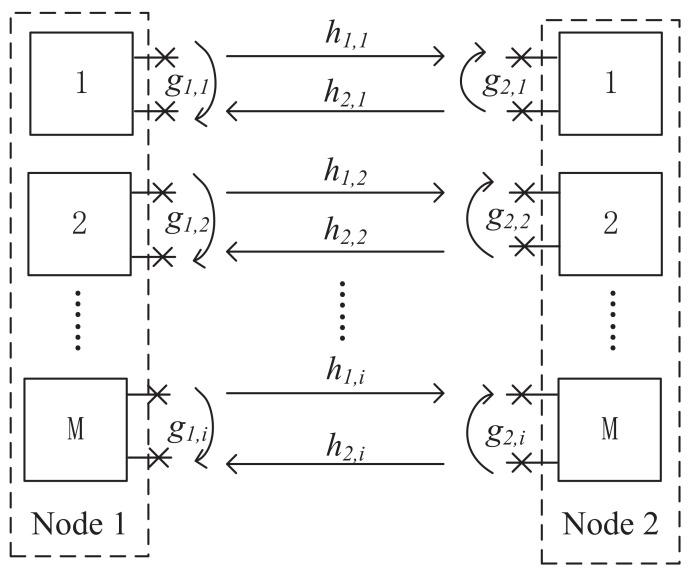
A full-duplex two-way communication system.

**Figure 2 entropy-24-00537-f002:**
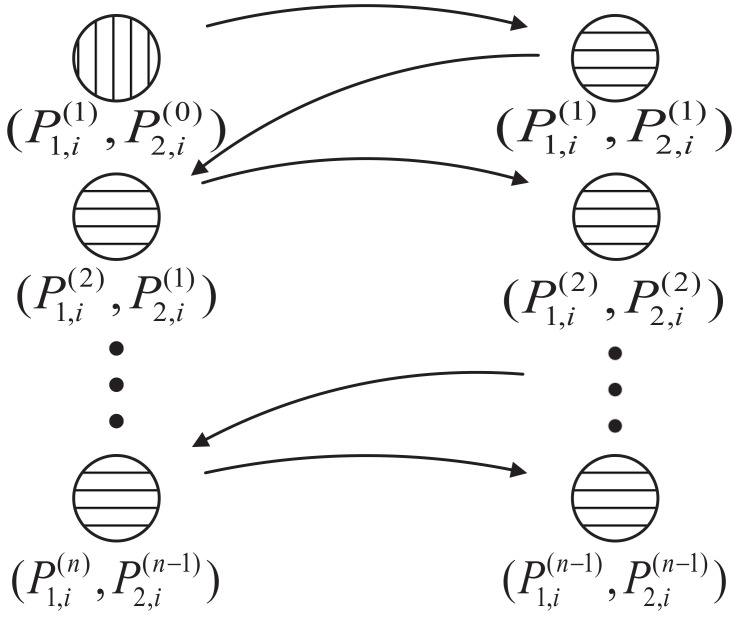
The iterative process of Algorithm 2.

**Figure 3 entropy-24-00537-f003:**
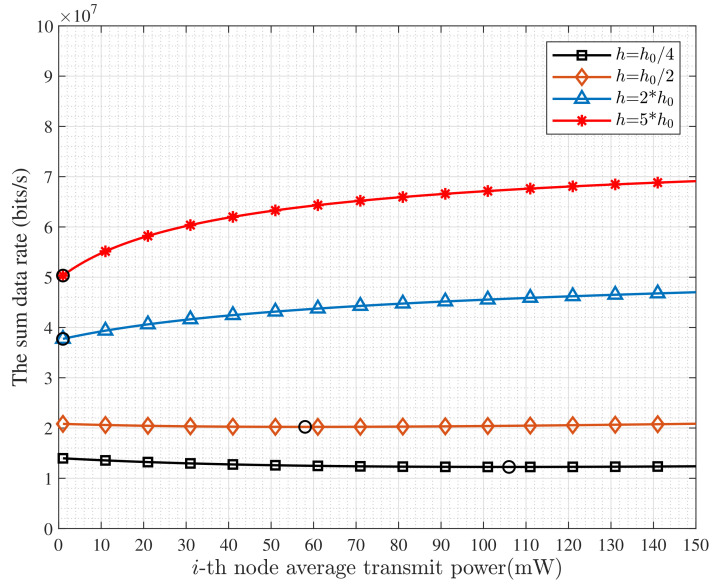
The $R_{s}$ versus power of $N_{1}$ under different *h*.

**Figure 4 entropy-24-00537-f004:**
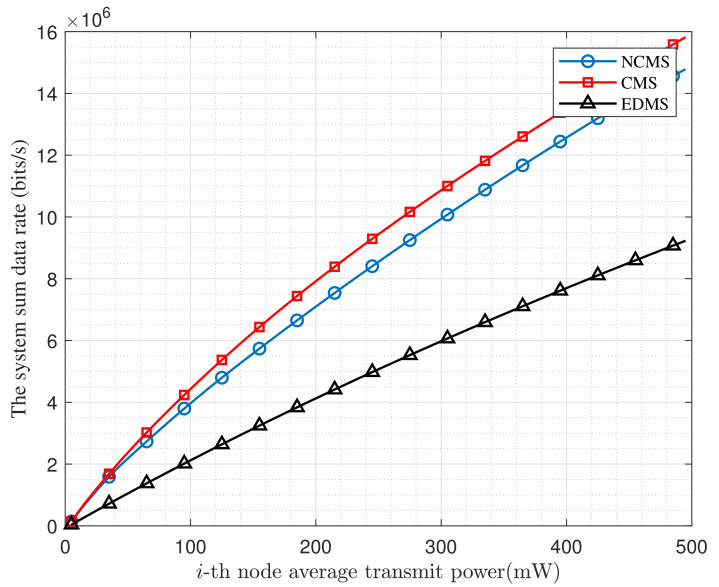
The $R_{m}$ under different conditions.

**Figure 5 entropy-24-00537-f005:**
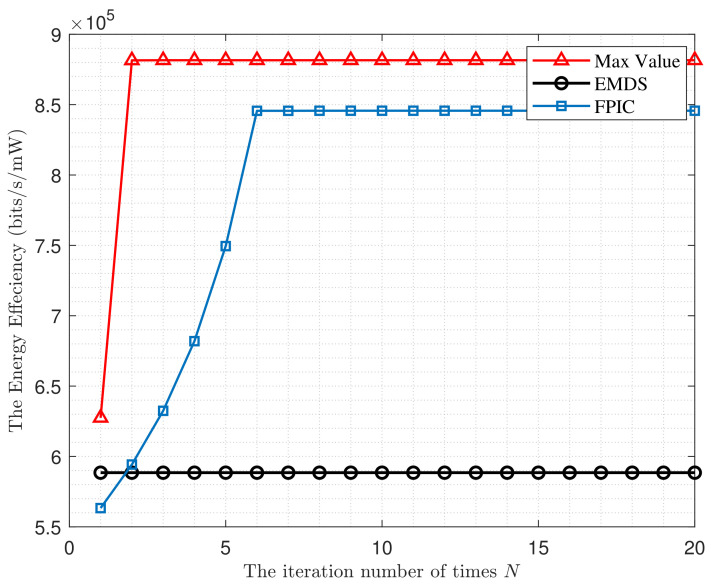
The $\eta_{m}$ under different conditions versus iteration number of times *N*.

**Figure 6 entropy-24-00537-f006:**
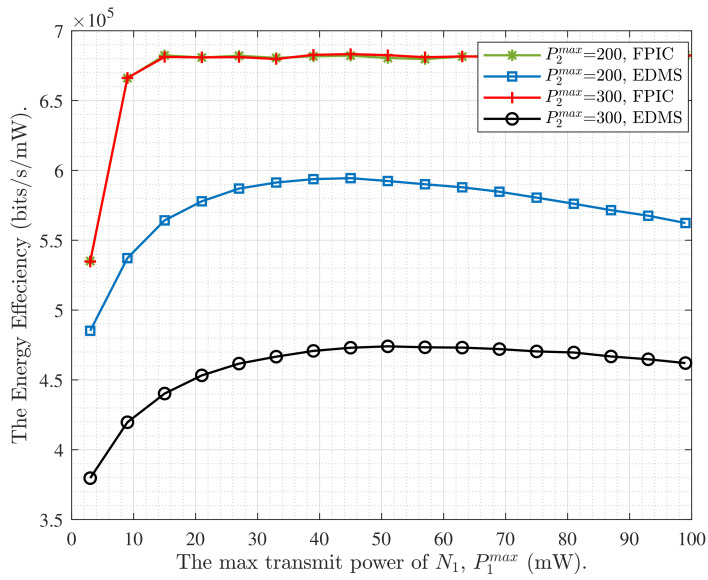
The $\eta_{m}$ versus power of $N_{1}$ under different $P_{2}^{\max}$.

## Data Availability

Not applicable.

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
