# Peer review of "The Optimal Power Allocation for Sum Rate and Energy Efficiency of Full-Duplex Two-Way Communication Network"

_entropy, 2022, doi:10.3390/e24040537_

Round 1
Reviewer 1 Report
The authors presented a power allocation problem in a full-duplex two way communication network over an OFDM channel for improving the sum data rate and energy efficiency. Moreover the authors showed that their method can significantly improve the sum rate and energy efficiency in both single-carrier and multi-subcarrier systems. The manuscript maybe interesting for the readership of MDPI research articles. The reviewer has the following comments.
(1) The current references are not enough to locate this work in state of the art. There are similar works published in MDPI Journals (Electronics, Sensors, Entropy etc). Include latest works to improve the section of literature review.
(2) It would be better if authors could compare their method with the existing methods of improving sum rate and energy in full-duplex systems. Add some discussion how much this method is efficient compared to the existing methods.
(3) Add a proper reference for Figure 1 if it does not belong to authors own work.
(4) Finally improve the conclusion and abstract by summarizing all the contributions of this work.
Author Response
Response to Reviewer 1 Comments
Thank you for your comments on our manuscript entitled "Improving Sum Rate and Energy Efficiency of Full-Duplex Two-Way Communication Network by Power Allocation" (entropy-1649231). Those comments are very helpful for revising and improving our paper, and also have the important guiding significance for other research. We have studied the comments carefully and made corrections which we hope meet with approval. The main corrections are in the manuscript and the responds to your comments are as follows (the replies are highlighted in red ).
Point 1: The current references are not enough to locate this work in state of the art. There are similar works published in MDPI Journals (Electronics, Sensors, Entropy etc). Include latest works to improve the section of literature review.
Response 1: We have added similar work on full duplex and ofdm to the introduction. There is added “ The authors of [14] proposed two-step method in which the power is optimally assigned to each subcarrier firstly, and then the assigned power is allocated to the two nodes at each subcarrier. Resource allocation in a full-duplex OFDMA wireless network had been discussed by using matching theory in [15]”.
Point 2: It would be better if authors could compare their method with the existing methods of improving sum rate and energy in full-duplex systems. Add some discussion how much this method is efficient compared to the existing methods.
Response 2: Thank you for pointing this out. We agree that more studies would be useful to enhanced the details. But much of the current works on FD and OFDM are based on relay, and many studies consider scenarios where the entire system has a common constraint, while our work focuses on considering different nodes with independent constraints. We believe that nodes with independent constraints would be more relevant to practical application scenarios. If we consider the point-to-point transmission scheme for comparison, then the result is clear that the multi-subcarrier transmission mode with OFDM will have a clear advantage. This makes us unable to do a side-by-side comparison with other schemes very well.
Point 3: Add a proper reference for Figure 1 if it does not belong to authors own work.
Response 3: We regret the inconvenience caused by Figure 1, and we have adjusted the sector descriptions for Figure 1 in order for the reader to better understand the system model in our work.
Point 4: Finally improve the conclusion and abstract by summarizing all the contributions of this work.
Response 4: Thanks to your suggestion, we have reorganized the summary of contributions in the abstract and conclusion to highlight our contributions in solving for the optimal transmit power of the system.

Reviewer 2 Report
The title is very confusing.
The introduction is very general. For less advanced readers, it may be incomprehensible.
Some of the abbreviations used are undefined.
Chapter two begins with a drawing. There is no introduction.
The work is about simulation. No scientific purpose and no practical purpose. In fact, it is not clear from all the work whether the goals have been achieved. Even if you admit that it did, the conclusions are too obvious. The conclusions are very general and do not confirm the achievement of the aim of the work. The presentation of the conclusions did not require the research and analyzes shown in the paper.
I do not understand why the work proposed just such and not other algorithms.
While reading the work, I noticed that there were few references to literature.
The results presented are quite random. The study does not present the methodological concept of the study. There is no experiment plan. There is no scientific inquiry methodology.
What is the scientific relevance of the results obtained? The authors conduct a fairly random theoretical and simulation analysis. And what will happen in practice? What is the statistical significance of the obtained results?
Author Response
Response to Reviewer 2 Comments
Thank you for your comments on our manuscript entitled "Improving Sum Rate and Energy Efficiency of Full-Duplex Two-Way Communication Network by Power Allocation" (entropy-1649231). Those comments are very helpful for revising and improving our paper, and also have the important guiding significance for other research. We have studied the comments carefully and made corrections which we hope meet with approval. The main corrections are in the manuscript and the responds to your comments are as follows (the replies are highlighted in red ).
Point 1: The title is very confusing.
Response 1: We regret that the title of our paper caused confusion for your reading and understanding, and we have changed our title to highlight our contribution to the search for the optimal transmit power of the nodes. The title has been changed to “The Optimal Power Allocation for Sum Rate and Energy Efficiency of Full-Duplex Two-Way Communication Network”.
Point 2: The introduction is very general. For less advanced readers, it may be incomprehensible.
I do not understand why the work proposed just such and not other algorithms. While reading the work, I noticed that there were few references to literature.
Response 2: Thank you for pointing this out. We agree that more studies would be useful to enhanced the details. But much of the current work on FD and OFDM are based on relay, or many studies consider scenarios where the entire system has a common constraint . We have added similar work on full duplex and ofdm to the introduction. There is added “ The authors of [14] proposed two-step method in which the power is optimally assigned to each subcarrier firstly, and then the assigned power is allocated to the two nodes at each subcarrier. Resource allocation in a full-duplex OFDMA wireless network had been discussed by using matching theory in [15]”.
Point 3: Some of the abbreviations used are undefined.
Response 3: Thank you for your comments and we feel sorry for our carelessness. We have defined the abbreviations that were not previously defined.
Point 4: Chapter two begins with a drawing. There is no introduction.
Response 4: We regret the inconvenience caused by Figure 1, and we have adjusted the sector descriptions for Figure 1 in order for the readersto better understand the system model in our work.
Point 5: The work is about simulation. No scientific purpose and no practical purpose. In fact, it is not clear from all the work whether the goals have been achieved. Even if you admit that it did, the conclusions are too obvious. The conclusions are very general and do not confirm the achievement of the aim of the work. The presentation of the conclusions did not require the research and analyzes shown in the paper.
Response 5: We regret that our research has caused yourconfusion. Our work is not focused on simulation, we focus to obtain a pair of optimal transmit power to improve the sum data rate and energy efficiency of a FDTW system based on OFDM channel. In this FDTW system, the nodes are more tightly coupled in terms of power, which makes the self-interference of the system more complicated. Hence, it is more difficult to efficiently allocate the power such that a decent sum data rate and energy efficiency can be achieved. We first formulate the optimal problem of selecting optimal power allocation strategy to maximize sum data rate and energy efficiency in a FDTW communication system based on OFDM channel, and each transmit node should satisfy the different power constraints. Then, the fractional programming and iteration algorithm are applied to get a suboptimal transmit power strategy. In particular, the proposed algorithms can approach the global optimal solution of transmit power for nodes and significantly improve the sum data rate and energy efficiency in the FDTW system with an OFDM channel.
Point 6: The results presented are quite random. The study does not present the methodological concept of the study. There is no experiment plan. There is no scientific inquiry methodology.
Response 6: In our work, although the nodes are directly communicating through a random wireless channel, the channel is a standard Rayleigh channel. Also our results do not represent the results of a particular random communication, on the contrary, the simulation results are the average results of multiple random delivery process satisfying the same Rayleigh channel conditions. This process was repeated at least 5000 times, so the results are representative, accurate and trustworthy.
Point 7: What is the scientific relevance of the results obtained? The authors conduct a fairly random theoretical and simulation analysis. And what will happen in practice? What is the statistical significance of the obtained results?
Response 7: The characteristics of full-duplex two-way communication will inevitably bring self-interference to the system, while the system uses multi-carrier transmission when it improves the system performance, but it makes the transmission power coupling complex between different nodes in the communication system. Therefore, solving for the optimal transmit power of the system in full-duplex two-way communication is a difficult problem. Our proposed algorithm can approximate the optimal solution of the system and the obtained optimal transmit power is viable for improving the transmission rate and energy efficiency of the system. In the actual scenario, the channels of the two communicating parties are random, but still conform to the Rayleigh distribution, and this distribution can be obtained by other technical means. In our simulation, the chance of a single random channel is overcome by a sufficient number of repeated trials, and the random channels in these repeated trials all obey the same distribution. This is also consistent with the random delivery process of the system, where the system is changing at any time, but has uniform distribution properties over time.

Reviewer 3 Report
This paper is a study of network performance and power efficiency through power allocation in a full-duplex two-way network.
This paper was generally well written, and the comments are as follows.
-In line 20,26,28 abbreviations are not declared properly.
-In line 75 of the system model a single variable ‘M’ is used for defining parallel channel, subcarriers and signal transmitted from node.
-In line 103,119,131,146,166 spellings are incorrect.
-In line 119 of the section 3.2, the variable M=1, is not clear about the parameter it is indicating.
-In line 131, “the node can choose a best value of transmit power to balance power and the transmit power of self-interference” the statement need to be more precise about the power balance.
-Exponent used for transmit power in Fig.2, line 141 and Algorithm 2 are not consistent.
-Contradiction in EDMS and EMDS in line 182 and Fig.4; EMDS is not defined
Author Response
Response to Reviewer 3 Comments
Thank you for your comments on our manuscript entitled "Improving Sum Rate and Energy Efficiency of Full-Duplex Two-Way Communication Network by Power Allocation" (entropy-1649231). Those comments are very helpful for revising and improving our paper, and also have the important guiding significance for other research. We have studied the comments carefully and made corrections which we hope meet with approval. The main corrections are in the manuscript and the responds to your comments are as follows (the replies are highlighted in red ).
Point 1: In line 20,26,28 abbreviations are not declared properly.
Response 1: Thanks for your careful checks. Based on your comments, we have made the corrections to make the unit harmonized within the whole manuscript. We have added a description of the abbreviation SIC in line 26, and the abbreviation of FD in line 20, 28 was introduced in abstract.
Point 2: In line 75 of the system model a single variable ‘M’ is used for defining parallel channel, subcarriers and signal transmitted from node.
In line 119 of the section 3.2, the variable M=1, is not clear about the parameter it is indicating.
Response 2: We have adapted the model introduction section by declaring that the number of carriers will be m=1 and m>1 for two different cases as “There are M parallel channels in the system, each of which uses a dedicated OFDM subcarrier with bandwidth W Hz.” Besides, when M = 1 the system is a single carrier communication system with only one channel, when M > 1 the system becomes a multi-subcarrier system with M subcarriers and each subcarrier is perfectly orthogonal to other subcarriers without inter-subcarrier interference.’’
Point 3: In line 103,119,131,146,166 spellings are incorrect.
Response 3: We have fixed the spelling errors in line 103,119,131,146,166.
Point 4: In line 131, “the node can choose a best value of transmit power to balance power and the transmit power of self-interference” the statement need to be more precise about the power balance.
Response 4: To make it easy to understand, we changed the statement as: “Nodes can choose an appropriate transmit power to achieve a balance between higher transmit data rate and stronger self-interference.”
Point 5: Exponent used for transmit power in Fig.2, line 141 and Algorithm 2 are not consistent.
Response 5: After careful checking and verification, we unified the description in Fig.2, line 141 and Algorithm 2 as |P1,i(n)-P1,i(n-1)| and |P2,i(n)-P2,i(n-1)|.
Point 6: Contradiction in EDMS and EMDS in line 182 and Fig.4; EMDS is not defined
Response 6: At the beginning of Chapter 5, the paper has introduced EDMS, and to help the reader understand it, we have made a reminder in line 184. There is added “Next, we compare the energy efficiency of FPIC scheme to that EDMS scheme in Fig. 5”

Round 2
Reviewer 3 Report
The authors have properly revised based on all comments.
Author Response
Thank you very much for your evaluation and comments on our paper.